# Sea lice (Lepeophtherius salmonis) detection and quantification around aquaculture installations using environmental DNA

**Adriana Krolicka**[1], **Mari Mæland Nilsen**[1¤], **Brian Klitgaard Hansen**[2☯], **Magnus Wulf Jacobsen**[2☯], **Fiona Provan**[1], **Thierry Baussant**[1] *

**1** Norwegian Research Centre AS (NORCE), Stavanger, Norway, **2** Danish Technical University, Section for Marine Living Resources, Silkeborg, Denmark

☯ These authors contributed equally to this work.
¤ Current address: University of Stavanger, Stavanger, Norway
* thba@norceresearch.no

**Data Availability Statement:** All data are available on Zenodo, please use following DOI 10.5281/

## Abstract

The naturally occurring ectoparasite salmon lice (*Lepeophtherirus salmonis*) poses a great challenge for the salmon farming industry, as well as for wild salmonids in the Northern hemisphere. To better control the infestation pressure and protect the production, there is a need to provide fish farmers with sensitive and efficient tools for rapid early detection and monitoring of the parasitic load. This can be achieved by targeting *L. salmonis* DNA in environmental samples. Here, we developed and tested a new *L. salmonis* specific DNA-based assay (qPCR assay) for detection and quantification from seawater samples using an analytical pipeline compatible with the Environmental Sample Processor (ESP) for autonomous water sample analysis of gene targets. Specificity of the L. salmonis qPCR assay was demonstrated through in-silico DNA analyses covering sequences of different *L. salmonis* isolates. Seawater was spiked with known numbers of nauplii and copepodite free-swimming (planktonic) stages of *L. salmonis* to investigate the relationship with the number of marker gene copies (MGC). Finally, field samples collected at different times of the year in the vicinity of a salmon production farm in Western Norway were analyzed for *L. salmonis* detection and quantification. The assay specificity was high and a high correlation between MGC and planktonic stages of *L. salmonis* was established in the laboratory conditions. In the field, *L. salmonis* DNA was consequently detected, but with MGC number below that expected for one copepodite or nauplii. We concluded that only *L. salmonis* tissue or eDNA residues were detected. This novel study opens for a fully automatized L. salmonis DNA quantification using ESP robotic to monitor the parasitic load, but challenges remain to exactly transfer information about eDNA quantities to decisions by the farmers and possible interventions.

## Introduction

The global aquaculture industry holds great promise as a provider of protein rich food to an increasing human population. However, disease, welfare and environmental issues are major

zenodo.6965644. High-throughput sequencing data are deposited under the following GeneBank BioSample accession number: SAMN29936430.

**Funding:** The authors received funding from Research Council of Norway (grant#267629) for this work. The funders had no role in study design, data collection and analysis, decision to publish, or preparation of the manuscript.

**Competing interests:** The authors have declared that no competing interests exist.

constraints to the development of this food sector, including the Atlantic salmon (*Salmo salar*) farming in Northern Europe [1]. Thus, to ensure a sustainable and continued growth of the global aquaculture production, there is a need for an increased focus on the development of effective approaches to detect, prevent and control aquaculture related diseases. One of the largest challenges to the global salmon farming industry is infestations by sea lice: primarily *Lepeophtheirus salmonis salmonis* (Krøyer, 1837) and *Lepeophtheirus salmonis oncorhynchii* (Skern-Mauritzen, Torrissen and Glover, 2014) in the Northern Hemisphere, although *Caligus elongatus (*van Nordmann, 1832) and *Caligus rogercresseyi* (Boxshall and Bravo, 2000) also has some impact, [2]. These parasitic copepods cause significant economic losses for the industry every year through decreased fish quality and reduced growth, treatment costs, secondary infections, stress and fish mortality [2–4]. *L. salmonis* is very specific with respect to the host and infect only *Salmonids* (mostly affecting the three salmonid genera *Salmo*, *Salvelinus*, and *Oncorhynchus*). *C. elongatus* is considered a generalist parasite and may infect a large variety of teleosts [5, 6], although it has been demonstrated that lumpfish (a cleaner fish) is strongly pre-ferred as a host [6, 7]. Medicated feed and bath medicines have traditionally been used to treat infestations, however, resistance to some of these treatments is becoming increasingly wide-spread (reviewed by Aaen and co-workers [8]). It has also been shown that several of the phar-maceutical agents currently used to control sea lice can have non-intended negative impacts on non-target species, such as other crustaceans, when released into the environment [9, 10]. *L. salmonis* is commonly present in the natural environment both in the Pacific and Atlantic oceans and feeds on mucus, epidermal tissue, and blood on host salmonid species. Thus, the proliferation of salmon lice in intensive salmon farming does not only impose huge costs on the aquaculture industry itself but can also lead to significantly increased mortality for wild salmon, sea trout (*Salmo trutta*) and anadromous arctic char (*Salvelinus alpinus*) in coastal regions [11, 12]. The life cycle of *L. salmonis* encompasses eight life stages, excluding the egg stage [13, 14]. The first three stages (nauplii I, nauplii II and copepodite) are free-swimming, non-parasitic stages where the larvae are passively drifted horizontally in the water column. However, it has been shown that the larvae can adjust their vertical position in the water col-umn in response to several environmental factors, such as light, salinity and temperature [15–19]. The planktonic stages, mainly the nauplii stages [6], must survive on their fat reserves until they find a suitable host and moult into the first parasitic stage; chalimus I (reviewed by Boxaspen [20]). Then chalimus I, after successful parasitic infestation, moult into the chalimus II, pre-adult I and II, and end in the final adult stage. The progression of the sea lice life-cycle varies depending on the temperature; however, at 10˚C, the time from fertilization of the egg to mature adult is from 38 to 40 days for males and from 44 to 52 days for females [13, 21].

As *L. salmonis* can be a threat to the health and welfare of both farmed fish and wild fish, the density of these organisms in the marine environment must be closely monitored, espe-cially in areas close to aquaculture sites, and treatments must be effectuated when needed. At present, monitoring of *L. salmonis* often involves manual inspection and counting of sea lice on farmed fish regularly. In several countries, this is set as a requirement by the government. Norwegian government regulations require, for example, a reduction of the *L. salmonis* burden if the average abundance exceeds 0.5 adult female parasite per fish evaluated during non-migration seasons (0.2 adult female parasite during the critical six-week spring season) every week or every second week. Norwegian regulation only requires that *L. salmonis* be monitored on a regular basis [6, 22]. The manual inspection of a certain number of fish per pen every week or every second week is a labour intensive and costly approach, it also imposes significant handling stress for the fish. The traditional approach is also only focusing on the parasitic stages of *L. salmonis* and is unable to detect and quantify the abundance of the first free-swim-ming life stages of *L. salmonis*. In addition, as only a limited number of fish are evaluated in

each pen, the numbers may not be representative for the total abundance level of *L. salmonis* in the surrounding environment. It has also recently been shown that Canadian salmon farming companies are regularly under-reporting the number of lice on their fish, most likely to avoid expensive delousing treatments [23]. Clearly, there is a need for a more cost-efficient, accurate, and less intrusive method for monitoring this species at any life stages to identify farm localities with high salmon lice density, and thereby high infestation pressure, at an early stage. This could allow fish farmers to take actions before the fish is infested, thereby limiting the negative effects on the fish and the environment. An alternative control strategy to the standard sea lice monitoring could consist in an effective non-disruptive detection of sea lice free-swimming (planktonic) load from environmental samples collected around fish farms with the help of molecular sensing and a robotic platform.

Analysis of environmental DNA (eDNA) is promising method for rapid and non-disruptive species detection in the aquatic environment (reviewed by Senapati and co-workers [24]), and might therefore be used as a management tool to quickly assess the *L. salmonis* parasite load in the marine environment. eDNA has been defined as DNA extracted directly from an environmental sample without any physical collection or visual signs of the biological source material and hence eDNA can originate from cells shed into the environment from larger organisms through e.g. excrements, epidermal mucus, gametes and saliva [25, 26]. However, others define it in its generic sense encompassing the DNA of all organisms present in environmental samples [27]. According to this definition collected eDNA also includes DNA of whole microorganisms, such as algae, bacteria and planktonic stages of living organisms in addition to DNA from non-whole organisms located both within (intracellular eDNA) or outside cell membranes (extracellular eDNA) [28]. The eDNA sampling procedure involves water collection, normally by filtration though a micro-pore filter and subsequent DNA analysis [25, 26]. By collecting seawater samples it is possible to detect and quantify species-specific DNA using molecular techniques and thereby evaluate the distribution and potentially the abundance of a species [29, 30]. So far, limited attention has been directed towards the detection of eDNA from crustaceans in the marine environment [31] and on the absolute (through quantitative PCR -qPCR) or relative quantification (though metabarcoding) of eDNA in marine samples. However, several recent studies have suggested that the eDNA concentration can reflect the local biomass of fishes in marine environments [32, 33]. Further, it was recently shown that it is possible to identify eDNA from various aquaculture pathogens [34–36], including *L. salmonis*, using eDNA metabarcoding (multi-species approach) of the 18S rRNA region in a mesocosm setting. The cost of eDNA metabarcoding and the complexity of the analysis might limit the use of the eDNA metabarcoding approach in a regulatory monitoring program. In contrast, qPCR allows faster analysis and is generally more cost-efficient if there are only a few target pathogen species or if the species are from various phyla or kingdoms, which often is the case for pathogens in aquaculture. Quantitative PCR also generally provides a better quantitative measure of eDNA fragments in seawater samples than a metabarcoding approach [34]. Finally, technological advances allow for qPCR analysis to be performed on-site [37] or even autonomously using so-called 'ecogenomic' sensors allowing for data in near real-time [38]. An example of such 'ecogenomic' sensor is the Environmental Sample Processor (ESP), which is essentially an underwater autonomous DNA laboratory that enables autonomous on-site water filtration, DNA extraction, qPCR analysis, and remote reporting, providing near real-time information about the occurrence and concentration of DNA targets within a few hours from initiation of sampling [39, 40].

The goal of this study was to design, validate and evaluate a new qPCR assay for detection and quantification of L. salmonis eDNA ("eDNA" in the sense defined by Pawlowski and co-workers [27]) in the marine environment. To validate and evaluate the assay we tested its

specificity and performance in silico and in vitro. We further investigated the quantitative aspects using spiking experiments to evaluate the relationship between gene copy number and the number of nauplii and copepodite individuals of L. salmonis. Finally, we tested and validated the assay functionality in the field, adapting the assay for use on a 2nd generation ESP [40–42].

## Materials and methods

### Ethics statement

The research presented only involved collection and analysis of water samples for eDNA, and manipulation of *L. salmonis* individuals for spiking experiments. NORCE is following the Norwegian animal welfare regulation regulated by the Norwegian Food Safety Authority (NFSA). NORCE is registered as a research facility in accordance with the NFSA and Use of Laboratory Animals (132-NORCE Mekjarvik). The personnel undergo a mandatory course organized by NFSA to assure the welfare of animals prior to use in research. However, experimental work with sea lice does not require approval by this authority. The infestation experiments of *Atlantic salmon* with *L. salmonis* were carried out at and by Skretting ARC who received approval from NFSA. Fish were housed for use in further research and animals were not sacrificed by the authors.

### Assay design

The Basic Local Alignment Search Tool (BLAST) (NCBI) was used to collect representative mitochondrial DNA (mtDNA) or nuclear ribosomal DNA sequences of members of the *Caligidae* family with sequences of *L. salmonis*. Subsequently, the obtained sequences were aligned using the MEGA7 software [43]. Based on the constructed alignment, the appropriate regions were identified: low in intra-species variation but highly divergent to homolog sequence form closely related species. Targeting the identified regions, qPCR assays primers and probes were chosen using PrimerQuest tool (Integrated DNA Technologies, Coralville, Iowa, USA) that incorporates Primer3 software (version 2.2.3) [44]. Three qPCR assays were designed, among them, two assays specific to mitochondrial DNA (mtDNA) and one to 18S nuclear ribosomal DNA (S2 Table). Finally, the Basic Local Alignment Search Tool (BLAST) (NCBI) and Primer-BLAST was used to do in silico analysis and select the most optimal set of oligonucleotides. To ensure specificity, primer pairs had at least 2 total mismatches to closely related non-target species, including at least 2 mismatches within the last 5 bps at the 3' end.

### Assay validation

**Limit of detection (LOD) and quantification (LOQ).**   LOD and LOQ were determined from a 11-point standard replicate curve, Standard curves were generated by serial dilution 1:5 (starting from the concentration of 550000 gene copies number), with eight technical replicates at each concentration. LOD was defined as the lowest concentration at which 95% of the technical replicates exhibited positive amplification. LOQ was determined at the lowest concentration at which the relative standard deviation of back-calculated concentrations was <25%.

**Assessing the relationship between number of individuals and gene copy number (spiking experiments).**   A stock of living individuals of *L. salmonis* individuals of two life stages (nauplii and copepodite) was purchased from the Industrial and Aquatic Research facility iLab in Bergen. Upon arrival to our facility 1 to 10 nauplii and copepodites, in 10 replicates, were placed in separate Eppendorf tubes and kept at -80˚C. To mimic realistic processing of seawater samples for eDNA detection (to include losses of DNA by extraction), but at the same time to examine the relationship between the number of individuals of *L. salmonis* and MGC

number, the following actions were undertaken. Sand-filtered seawater (200 ml) collected from 80 meters depth in Byfjorden (59.02283N, 5.62376E) by the NORCE facility was vacuum-filtered onto a 25 mm diameter 0.22 μm pore size Durapore filter (Merck Millipore, Burlington, Massachusetts, USA) before each filter was spiked with exactly 1, 2, 3, 5 and 10 either *L. salmonis* nauplii or copepodites per filter. The filters were preserved at -80˚C prior to DNA extraction and analysis.

**Infestation experiments—a preliminary evaluation for eDNA detection of sea lice.** To evaluate whether *L. salmonis* eDNA could be detected in seawater samples using the developed *L. salmonis* qPCR assay, seawater samples were collected from aquarium tanks with a high density of *L. salmonis* from bath infestation experiments performed at Skretting ARC in 2017. Briefly, the bath infestation experiments were performed by stopping the water flow in the fish tanks (approximately 30 fish in each tank) and adding a specific number of sea lice copepodites (Aquatic Research facility iLab, Bergen https://www.uib.no/forskning/74634/ilab) in the tanks with *Atlantic salmon*. The water flow was then resumed after two hours. Seawater samples was collected from the surface water approximately two weeks later. The seawater samples were collected into autoclaved glass bottles and were kept on ice during transport (approximately 1 hour). Immediately after arriving at the NORCE facility in Mekjarvik, the samples were vacuum filtered onto a 0.22 μm pore size Durapore filter (Merck Millipore, Burlington, Massachusetts, USA). Four samples of various volume (from 2 to 6 liters per sample) from tanks with sea lice-infested fish were filtered. Seawater samples (ranging from 4–5 liters) were also collected using the same protocol as above from a tank with no infested fish. This tank was placed outdoors with seawater directly pumped into the tank from 80 meters depth.

**Field samples.** Sampling around fish pens—Field sampling was conducted at an Atlantic salmon farm in the Western part of Norway (Kvitsøy, 59.05714N, 5.44000E). 27, 29 or 24 seawater samples (in total 80 samples, ~1L each) were collected respectively in October 2019, May 2020, and September 2020 at 3 depths (1, 5 and 10 meters) and at 4 locations of various distances from the fish pen nets (S1 Fig). A Cole-Parmer © Masterflex portable sampling pump was used to directly filter the samples through a Durapore filter with a 0.22 μm pore size and a diameter of 47 mm (Merck Millipore, Burlington, Massachusetts, USA). The filtered samples were brought to the laboratory within 3–4 hours where they were stored (-80˚C).

Sampling in a region of low aquaculture density—Thirty-nine seawater samples from 20 localities in the Oslofjord were collected in November 2018 (S1 Table) and included to evaluate the background levels of *L. salmonis* eDNA in seawater samples. The specific area is characterized by few fish farms and the abundance of the target species is thus expected to be relatively low (https://kart.fiskeridir.no/akva). Overall, 1 L of seawater samples was collected from 3–4 m depth (S1 Table) and filtered using a MF-Millipore™ Membrane Filter, with 0.22 μm pore size and a diameter 47 mm (Merck Millipore, Burlington, Massachusetts, USA).

## Sample preparation and analytical work

**DNA extraction.** DNA extraction of the filtered seawater samples was performed using a method that mimics the ESP DNA extraction workflow [42] with slight modifications as described in [45]. After extraction, the concentration of DNA in each sample was measured using the Qubit dsDNA HS (High Sensitivity) Kit (Thermo Fisher Scientific, Carlsbad, California, USA) before the samples were stored at -20˚C.

**qPCR analyses.** The qPCR analyses of all collected experimental and field samples were performed on a StepOnePlus instrument (Thermo Fisher Scientific, Carlsbad, California, USA). For the construction of the standard curves, synthetic target gene amplicons were used (gBlocks®, Integrated DNA Technologies, Coralville, Iowa, USA). For the standard dilution

series, a synthesized DNA fragments of *L. salmonis* 16S rRNA mitochondrial gene (GenBank ID EU288200), nuclear 18S rRNA (GenBank ID AF208263) or CO1 (GenBank ID LT630766.1) were utilized. The final concentration of primers and the probes (S2 Table) equaled 250 nM in each reaction for SL1 and SL2 assays. PCR thermal conditions were as follows: 20 s at 95˚C of initial denaturation, then 50 cycles of 20s at 95˚C and 30s at 60˚C. PCR reactions contained 10 μL 2x TaqPath™ qPCR Master Mix, CG (Thermo Fisher Scientific, Carlsbad, California, USA), 2μL template DNA and ddH$_2$O to a final volume of 20 μL. As a comparison, the samples were also analyzed using the MC qPCR assay [46] targeting region of CO1 gene of mt-DNA. For these PCR reactions, thermal conditions and the final concentration of primers and the probe (S2 Table) were as recommended by McBeath and co-workers [46]. Negative template control (NTC) reactions without any template DNA were carried out simultaneously on each plate. For all qPCR assays, PCR reactions were performed in duplicates. For all samples, assessment of potential PCR inhibition was evaluated with Internal Positive Control (IPC) amplification using TaqMan® Exogenous Internal Positive Control Reagents (Applied Biosystems, Foster City, CA, USA), 10 μL of 2X TaqPath qPCR Master Mix, CG, and 2 μL of the undiluted DNA extracts. This analysis was also performed on the StepOnePlus instrument (Thermo Fisher Scientific, Carlsbad, California, USA) according to the manufacturer's instructions, and using the same PCR thermal conditions as for the SL2 assay. Ct value (mean ± SD) 25.8 ± 0.20 was used to identify signatures of PCR inhibition in environmental samples. This was determined in PCR reactions for which DNA free water was used as a template.

**High throughput sequencing of amplicons.** The MiSeq illuminia sequencing platform was used to check the specificity *L. salmonids* -targeted qPCR assays by sequencing amplicons generated using SL2 qPCR assay (S1 Appendix).

## Implementing and testing the assays on the ESP

To assess the performance of the assays on the ESP to detect and quantify *L. salmonis*, we compared standard curves generated using the StepOnePlus instrument to standard curves generated on the analytical module, commonly known as the microfluidic block (MFB), on the ESP. The MFB is a module on the ESP, which is responsible of three processes i.e. microfluidic handling, DNA purification and qPCR analysis. The triplicate reactions performed on the ESP consisted of 6 μL DNA template, 6 μL assay mix consisting of primer and probes in 1×TE buffer and 18 μL mastermix consisting of 15 μL 2x TaqPath™ qPCR Master Mix, CG and 3 μL ddH$_2$O. The thermal profile and final primer and probe concentrations were identical to those on the StepOnePlus. The assay mix and mastermix were prepared in a dedicated PCR-free clean-lab facility and stored on the ESP at room temperature in closed containers wrapped in tinfoil to protect them against light. The standard template was fed into the system through an inlet tube into the MFB module and reactions were autonomously assembled by the MFB [42]. The same standard stock was used on both the bench StepOnePlus and ESP instrument for a direct comparison. After each reaction, the ESP decontaminated itself using household bleach (1% sodium hypochlorite) followed by rinsing with nuclease free ddH$_2$O. To assess potential contamination negative control reactions were analyzed before running standards. Further, to avoid influential carry-over contamination, standards were always run in sequential order from lowest to highest standard concentration (6×102–6×10$^5$ copies).

## Statistical analysis and illustration of results

GraphPad Prism version 5.0 was used (GraphPad Software, San Diego, California USA) to test for normal distribution of the data and further test for significant differences between the

number of MGCs of *L. salmonis* at individual depths. Graphics was prepared using either the same software or Excel Office 365.

## Results

### qPCR assays characterization

**In-silico analysis.** Three new qPCR assays were designed and tested (S1 Table). The qPCR assay targeting 18S rRNA (S1 Table) was discarded at an early stage from further evaluation due to low in silico specificity, this was also confirmed later by undesirable weak amplification in control samples without DNA target. The performance of the two other developed qPCR assays (SL1 and SL2 assay), targeting the 16S rRNA mitochondrial gene, were similar when tested in silico. The NCBI search by using following keywords [Lepeophtheirus salmonis isolate] AND [16S rRNA] resulted in 259 hits meaning that coverage of 16S rRNA *L. salmonids* by SL1 and SL2 is very high, namely 100% in the case of both assays when taking into the account individual oligonucleotides (Table 1). Primer-BLAST search against *Caligidae* revealed, taking into the account the number mismatches for forward and reverse primer, very low chance of amplification of a sister *Caligus* species (Fig 1), especially for the SL2 qPCR assay. In most of the cases there is a perfect match between *L. salmonis* hits and SL2 primers (88% hits for forward and 91% for reverse primer) (Fig 1) and probes (S2 Fig). Moreover, the novel SL2 assay targets both the Atlantic and Pacific *L. salmonis* 16S rRNA mitochondrial sequences ([GenBank id:EU288264-EU288330 and AY602770-AY602949 [47, 48], indicating that the assays can be successfully used for analyzing different populations (S2 Fig). The results of the in-vitro evaluation for the SL2 assay (S2 Table), for which delta fluorescence was slightly higher than for the SL1 assay, were analyzed later in detail (LOD and LQD, spiking experiments and field data).

**LOD and LQD of SL2 assay.** The analysis of LOD revealed that it was possible to detect concentrations down to one DNA copy (S3 Fig) and reliably quantify DNA concentrations as well down to one DNA copy (S4 Fig).

**Spiking experiments–molecular quantification of L. salmonis abundance.** The performance and the sensitivity of the SL2 and MC qPCR assays were assessed by quantifying the number of gene copies in seawater spiked with different numbers of *L. salmonis*. There was a strong linear and significant relationship (p<0.001) between number of copepodite and nauplii vs. the number of MGC for the SL2 assay (Fig 2). The correlation coefficient ($R^2$) between numbers of individuals of copepodite and nauplii vs. the number of MGC was 0.90 (n = 45) and 0.93 (n = 47), respectively. There was further a strong linear relationship between MGC determined using the MC assay and number of copepodite ($R^2$ = 0.91, n = 46) and nauplii ($R^2$ = 0.86, n = 48) individuals (Fig 2).

**SL2qPCR assay testing in the experimental setup with high density of L. salmonis.** To evaluate the performance of SL2 qPCR assay and evaluate if it is possible to detect eDNA directly from the seawater, a set of filtered seawater samples collected from tanks with *L. salmonis* infested fish were analyzed. Results show that the number of MGC per liter of filtered seawater was at least two orders of magnitude higher in tanks with sea lice-infested fish compared to the tank with non-infested fish, indicating that it is potentially possible to detect *L. salmonis* in seawater samples from the field (S5 Fig).

**L. salmonis eDNA quantity in field samples and SL2 assay specificity.** Sampling around fish pens -The undiluted DNA extracts from the field did not demonstrate signatures of PCR inhibition. This excludes the occurrence of false negatives due to the PCR inhibition. Using the SL2 qPCR assay, *L. salmonis* was detected in seawater samples at all four stations around the fish farm at Kvitsøy in May 2020 and September 2020. The numbers of MGC per 1mL of

**Table 1. SL2 and SL1 primers and probes specificity characteristics based on the in-silico evaluation using blastn.** The table includes *Lepeophtheirus* BLAST hits and the best non-*Lepeophtheirus* hits (on bold).

| Organism | Coverage | Number of Hits (Identities) | Description |
|---|---|---|---|
| **SL1 assay** | | | |
| **FORWARD PRIMER** | | | |
| Lepeophtheirus salmonis | Full | 262 (100%) | Lepeophtheirus salmonis 16S rRNA mitochondrial hits |
| Lepeophtheirus salmonis salmonis | Full | 1 | Lepeophtheirus salmonis salmonis 16S rRNA mitochondrial hits |
| Lepeophtheirus salmonis oncorhynchi | Full | 1 | Lepeophtheirus salmonis oncorhynchi 16S rRNA mitochondrial hits |
| **Caligus rogercresseyi (crustaceans)** | **18/21** | **8** | **Caligus rogercresseyi hits** |
| **REVERSE PRIMER** | | | |
| Lepeophtheirus salmonis | Full | 266 (100%) | Lepeophtheirus salmonis 16S rRNA mitochondrial hits |
| Lepeophtheirus salmonis salmonis | Full | 1 | Lepeophtheirus salmonis salmonis 16S rRNA mitochondrial hits |
| **Caligus rogercresseyi (crustaceans)** | **Full** | **7** | **Caligus rogercresseyi hits** |
| **PROBE** | | | |
| Lepeophtheirus salmonis | Full | 264 (100%) | Lepeophtheirus salmonis 16S rRNA mitochondrial hits |
| Lepeophtheirus salmonis salmonis | Full | 1 | Lepeophtheirus salmonis salmonis 16S rRNA mitochondrial hits |
| Lepeophtheirus salmonis oncorhynchi | 23/24 | 1 | Lepeophtheirus salmonis oncorhynchi 16S rRNA mitochondrial hits |
| **Pollicipes pollicipes (crustaceans)** | **16/24** | **1** | **Pollicipes pollicipes hits** |
| **SL2 assay** | | | |
| **FORWARD PRIMER** | | | |
| Lepeophtheirus salmonis | Full | 260 (100%) | Lepeophtheirus salmonis 16S rRNA mitochondrial hits |
| Lepeophtheirus salmonis salmonis | Full | 1 | Lepeophtheirus salmonis salmonis 16S rRNA mitochondrial hits |
| Lepeophtheirus salmonis oncorhynchi | Full | 1 | Lepeophtheirus salmonis oncorhynchi 16S rRNA mitochondrial hits |
| **Bactrocera dorsalis (flies)** | **17/23** | **1** | **Bactrocera dorsalis hits** |
| **REVERSE PRIMER** | | | |
| Lepeophtheirus salmonis | Full | 258 (100%) | Lepeophtheirus salmonis 16S rRNA mitochondrial hits |
| Lepeophtheirus salmonis salmonis | Full | 1 | Lepeophtheirus salmonis salmonis 16S rRNA mitochondrial hits |
| Lepeophtheirus salmonis oncorhynchi | Full | 1 | Lepeophtheirus salmonis oncorhynchi 16S rRNA mitochondrial hits |
| **Macrobrachium nipponense (crustaceans)** | **18/20** | **2** | **Macrobrachium nipponense hits** |
| **PROBE** | | | |
| Lepeophtheirus salmonis | Full | 250 (100%) | Lepeophtheirus salmonis 16S rRNA mitochondrial hits |
| Lepeophtheirus salmonis salmonis | Full | 1 | Lepeophtheirus salmonis salmonis 16S rRNA mitochondrial hits |
| Lepeophtheirus salmonis oncorhynchi | Full | 1 | Lepeophtheirus salmonis oncorhynchi 16S rRNA mitochondrial hits |
| Lepeophtheirus pollachius | 23/24 | 1 | Lepeophtheirus pollachius 16S rRNA mitochondrial hits |
| **Caligus rogercresseyi (crustaceans)** | **23/24** | **8** | **Caligus rogercresseyi hits** |

seawater were low. The highest concentration of eDNA targets was recorded in September at station 3 and was estimated to 136 copies per 1 mL of seawater ($1.4 \times 10^5$ per 1L). *L. salmonis* DNA was also detected (at the level of 0.5–3 copies per 1ml of seawater) in three samples from two stations (station 1. and 3.) collected in October 2019 (Fig 1). In the remaining samples collected in October no eDNA targets were detected. The number of MGC observed in the field samples was more than two orders of magnitude lower, than what observed for one individual (in the nauplii stage). A significant difference in the number of MGCs of *L. salmonis* at individual depths in September 2020 was observed (Kruskal-Wallis test, $p<0.05$), but not in May 2020 (Kruskal-Wallis test, $p>0.05$) (S6 Fig). In September the highest number of MGCs was observed in the upper layer, at the 1 m depth. The detection rate and estimated concentrations of sea lice found using the MC assay [46] was considerably lower for all samples. The difference in detection was particularly visible in May 2020. Here there was a positive detection for all samples and instrumental replicates using the SL2 qPCR assay while the MC qPCR

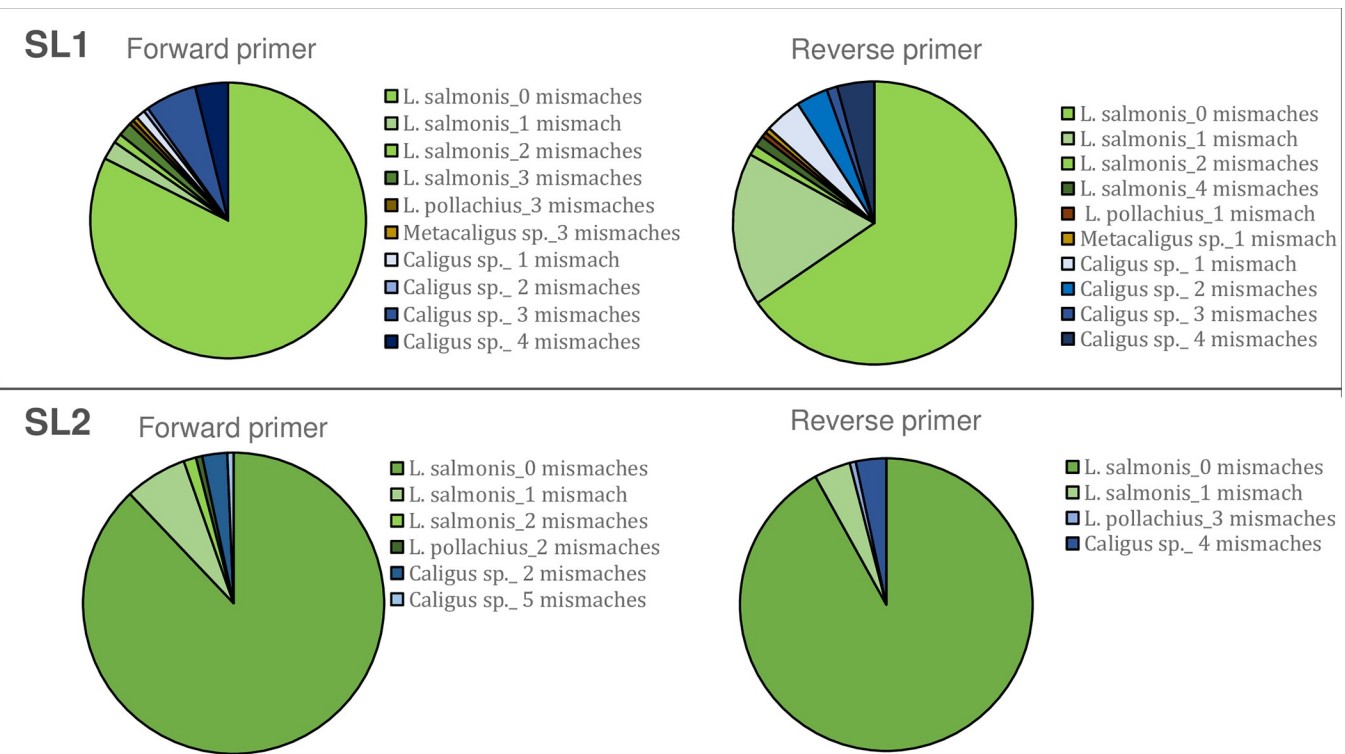

**Fig 1. Specificity of SL1 and SL2 assays, a result of Primer-BLAST search.** The figure includes hits up to 5 mismatches within the last 5 bps at the 3' end.

(CO1-based) qPCR assay [46] demonstrated much lower numbers of MGC and much higher variability, and in several cases lack of amplification where the SL2 assay would amplify.

To ensure that the quantity of DNA targets detected was only from *L. Salmonis*, high throughput sequencing was performed for all merged qPCR-based amplicons generated using

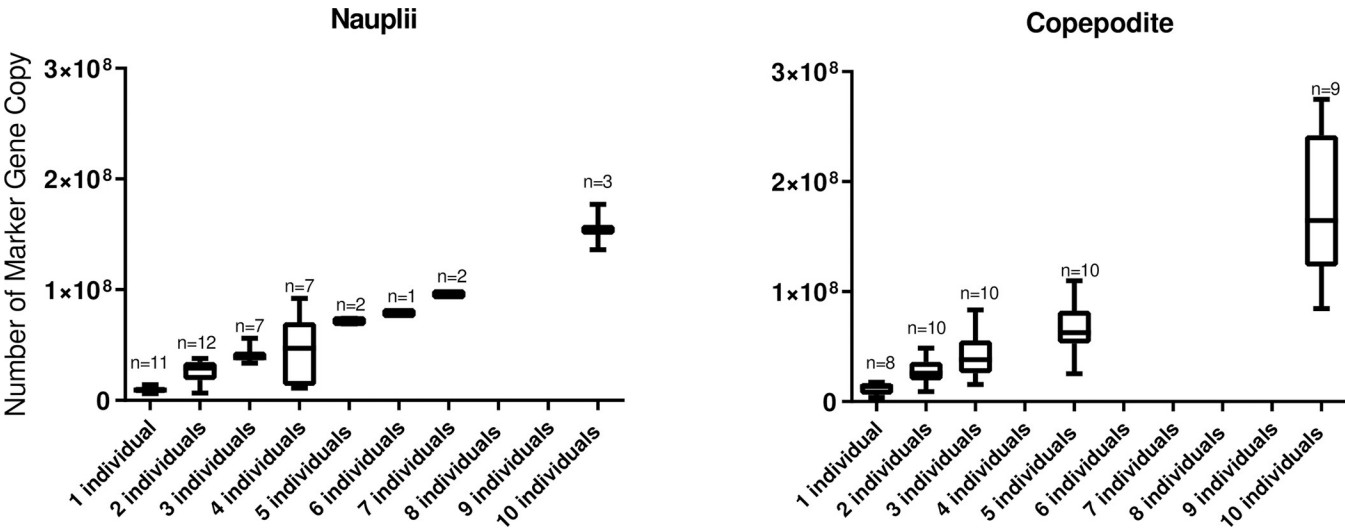

**Fig 2. SL2 qPCR assay and number of individuals.** Boxplot of the relationship between the number of MGC (Marker Gene Copy) and the number of nauplii and copepodite individuals per sample, n = number of samples analyzed.

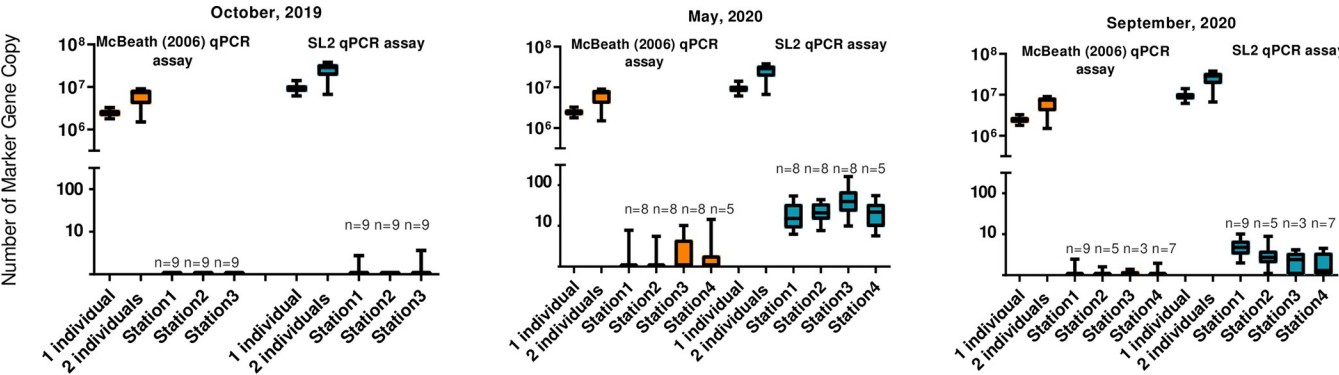

**Fig 3. Number of estimated MGC in field samples (per 1 mL of seawater) collected in October 2019, May 2020, and September 2020.** The field data include results obtained for samples collected from 1m, 5m, 10m depth determined by using the MC assay (orange) and the SL2 assay (blue). For the comparison, the total number of MGC corresponding to 1–2 individuals (nauplii stage) are also included in the graph. Sample size (n) is depicted for each analyzed station.

samples collected in May 2020. The results of this analysis showed that the SL2 qPCR assay had a very high specificity. Two OTUs were identified, wherein OTU1, 100% identical to *L. salmonis* mitochondrial sequence, strain IoA-00 (GenBank ID LT630766. 1), constituted 99% of the generated sequences. OTU2 was 97% identical to the same reference sequence.

Background levels of L. salmonis in Oslofjord- *L. salmonis* was not detected at all in 39 samples collected from the Oslofjord (S1 Table) via qPCR amplification using SL2 qPCR assay. The IPC did not reveal any PCR inhibition. This can indicate that there was no amplification from unintended targets.

## Performance of SL2 assay on the ESP

Overall, the assays performed quite well on the ESP when compared to benchtop setup. The SL2 assay had an efficiency of 109.2% with an $R^2$-value of 0.98 on the ESP. In comparison, the standard curve analyzed via the StepOnePlus qPCR instrument, using the same reaction stock, showed an efficiency of 103.7% with a $R^2$- value of 0.99. Similarly, the MC assay showed an efficiency of 101.9% with an $R^2$-value of 0.96 on the ESP and showed efficiency of 106.7% and a $R^2$- value of 0.99 on the StepOnePlus (S7 Fig).

## Discussion

Salmon lice is one of the most significant parasite threats to salmonid production in aquaculture and one of the greatest limiting production factors due to associated mortalities, as well as expenses related with the extensive and frequent treatments required. The issue has prompted the search for alternative adequate monitoring methods, ideally providing results in real time to be able to immediately introduce suitable actions to prevent large scale salmon lice invasion and infection. In a recent study [49] none of available methods passed the comparative test of salmon louse enumeration in plankton samples. These included visual-based -fluorescence microscopy and automated fluid imaging and molecular-based—droplet digital PCR (ddPCR), quantitative fraction PCR and qPCR. This suggests that a compromise among performance with precision, time used on the analysis and the costs has to be made when choosing a method. The results of qPCR analysis did not prove to be highly accurate and the ddPCR method performed better than qPCR [49]. The outcome of this kind of evaluation may be different when using the newly developed qPCR assays since the successful and accurate enumeration is highly depended on the assay. Based on the results of the analyses of the field samples,

the SL2 qPCR assay showed higher sensitivity than the MC qPCR assay [46]. The higher sensitivity of the novel SL2 qPCR assay may be related to the coverage of the MC qPCR assay in respect to *L. salmonidis* hits. This can indicate a need for the detailed examination of the MC qPCR assay if comes to the range of qPCR coverage based on the updated (in the past 14 years) GeneBank database. The higher detection rate of SL2 assay cannot be explained by lower specificity of the assay since the amplicons generated from the analyses of the field samples contained only salmon lice sequences. Given that both assays target regions within the mitochondrial genome they were expected to show similar results.

We detected *L. salmonis* eDNA at all four sampled locations around the salmon farm and from the three different sampled depth of 1, 5- and 10-meters. There were significant differences in the estimated *L. salmonis* MGC number at individual depths in September 2020, but not in May 2020. This possibly relates to different salinity, light and/or temperature conditions [50] in these two months. In September, the light penetration to deeper layers of seawater was weaker than in May. The more homogeneous light and temperature conditions in May could have resulted in a more homogeneous distribution of *L. salmonis* and their eDNA. Another explanation is that the *L. salmonis* population represented different developmental stages between the two analyzed months. It has been shown in other experimental studies that temperature does not influence the vertical distribution of copepodites in contrast to nauplii larvae for which vertical distribution is temperature dependent [51]. Since the spatial and temporal signal of eDNA is dependent heavily on the environment it is possible that vertical mixing of water masses and/or degradation rates was more of importance than sea lice distributions. Season, light conditions, and water temperature are among the most important factors that impact distribution and eDNA stability [52]. The light conditions were different in May than September, UV light intensity was stronger in May than in September, therefore it is possible for example that UV light impacted on faster eDNA decay in upper layer in May, therefore no significant differences in MGC between individual layers were observed. The available online data of salmon lice occurrence in the farm (https://www.barentswatch.no/fiskehelse, S8 Fig) demonstrates that in October 2019, there was a very low number of parasites found on the fish (<0.5 individuals). A similar situation was reported for September 2020 (0.7 individuals) while the highest number of parasites on fish were detected in May 2020 (0.9 individuals). This is in accordance with our results, which overall show low concentration of *L. salmonis* DNA in May and September 2019 and relatively higher *L. salmonis* concentrations in May 2020. This indicates a good correspondence between established monitoring practices and the eDNA-based measurements. In addition, our results are in accordance in some extend with the literature describing planktonic lice abundances (Norwegian coast and Central Norway) where approximately 1–5 individuals per m$^3$ were sampled in seawater around the fish pens [6]. Considering the concentration of sea lice eDNA observed per 1L of seawater and the estimated total DNA found per 1 individual (nauplii) this translates into 1–1.5 planktonic lice individuals per m$^3$ for our data from May. Although this number is based on eDNA targets and not entire organisms (because much smaller volume of water was collected than cubic meter), the estimated numbers of salmon lice are comparable to the results of traditional monitoring. Obviously, these findings need further confirmation by replicated analyses.

The higher sensitivity of the SL2 qPCR assay and specificity to *L. salmonis* can become a valuable tool for aquaculture monitoring. We obtained similar efficiency, but slightly lower correlation coefficients for both assays when compared to the StepOnePlus instrument. This demonstrates that both the MC assay [46] and the newly developed SL2 assays are compatible with the ESP technology, which makes it possible in the future to investigate the potential for autonomous on-site qPCR-based monitoring. The implementation of these two assays makes it possible in the future to investigate the potential for autonomous on-site monitoring of *L*.

*salmonis*. Furthermore, even though the *L. salmonis* qPCR assay was only validated in the laboratory on *L. salmonis* obtained from one population of *L. salmonis* from the Atlantic Ocean, *in silico* analysis strongly indicates that the new assay should work for *L. salmonis* populations from both the Pacific and Atlantic ocean. Recent studies demonstrated weak population genetic differentiation among *L. salmonis* sampled not only from geographically distinct regions but also between the Pacific and Atlantic oceans [47, 53,54], suggesting a very high level of gene flow due to its high dispersal potential either passively by ocean currents or while attached to its highly migratory hosts [54].

Compared to the traditional methods, which are based on manually counting adult stages on the fish, an eDNA approach potentially has several advantages; 1) the results are delivered fast and efficiently; 2) it can reduce the time and costs associated with the monitoring as the method does not require host collection and manual counting; 3) it can detect a variety of stages of *L. salmonis*, including the first free-swimming stages, allowing for more representative measurements of the abundance and a better method to map the spread of infection in time and space; 4) it can cover several depths and seasons, and thereby produce more comprehensive biological data for *L. salmonis* abundance; 5) the method is non-invasive for the fish and thus does not affect fish welfare. Furthermore, the qPCR assay was developed with the thought to be fully compatible with automatization on the ESP. This instrument enables autonomous on-site sampling, filtration and DNA analysis of seawater samples, and further real time streaming of results [55]. Using such a device would enable the farmers to obtain autonomous, frequent and rapid data on the presence of changes in the abundance of early stages of *L. salmonis* in the seawater. This would enable a rapid counteraction to mitigate the potential infection risk for the farmed fish. Should the ESP indicate a disease-free status; no actions are needed from the fish farmers. However, when the ESP reveals an increase in abundance (a problem), fish farmers will be able to take more immediate action to prevent an outbreak before fish get infected. This will most likely result in fewer fish being infected and less treatment needed, which could be beneficial in terms of fish welfare, economy, and environment impact. Another advantage of using an ESP is that several qPCR assays can be included in one device [40], allowing for detection and quantification of a range of target infectious or pathogenic species simultaneously. For example, for the fish farming industry it would be beneficial if they could monitor the presence and abundance of multiple pathogens or parasites, such as *L. salmonis*, the amoeba *Neoparamoeba perurans*, which causes amoebic gill disease [56] and the Salmon ISA virus which causes infectious salmon anemia [57] in the seawater column simultaneously. However, to use the eDNA approach for multiple diseases in aquaculture requires further development of protocols prior the implementation into robotic devices such as the ESP. For instance, better methods for concentrating eDNA and size fractionation depending on sizes would increase the chances of the capture and detection of disease causative agents.

One challenge with using the eDNA approach to monitor *L. salmonis* is to establish how eDNA copy number relates to biomass. Compared to unicellular species, absolute individual-level quantification is complicated in metazoans where biomass, instead of count data, is likely to show better correlation with DNA quantity [58]. Furthermore, there is still some uncertainty related to the degradation rate and the dispersal rate of eDNA in aquatic systems. Several studies have suggested that eDNA can be used quantitatively, but for relative rather than absolute quantification [59]. Experiments have shown a rapid degradation of eDNA in freshwater [26]. Fewer studies have been performed on the persistence of eDNA in seawater, Collins and co-workers [60] indicated that eDNA may be detected for around 2 days whereas according to Thomsen and co-workers [61] up for 7 days. Transport of eDNA within ecosystems could be a challenge in flowing waters and even more in marine environments. Nevertheless, the

degradation of eDNA in aquatic systems has been found to occur at a scale of days or weeks [60–62], rendering long-distance dispersal unlikely. This is particularly important in large open systems such as oceans, where sea currents could potentially transport eDNA over large distances. However, there might also be species-specific differences in DNA persistence [61] and this needs to be further investigated for *L. salmonis* eDNA. In environments where DNA often is present at low concentrations and/or is degraded, the greater number of mtDNA per cell than the nuclear DNA becomes especially important for its detection. Furthermore, due to the relatively rapid degradation of eDNA within seawater, it is important to use a small fragment size as an assay target as larger fragments will be less likely to persist long enough to allow species detection [63]. Our study demonstrates the great potential for applying eDNA towards current challenges in aquaculture and gives promise for faster and less time-consuming manner of early detection of incoming threats.

## Conclusion

We developed and analyzed a novel species-specific qPCR assay, which targets (e)DNA from the salmon fish parasite *L. salmonis*, and which is compatible with the ESP. The results can indicate that the SL2 qPCR assay can be used for reliable detection and quantification of *L. salmonis* eDNA in the water column. Thus, a DNA-based monitoring method that would not require host collection and manual counting, represents a relatively simple, non-intrusive and cost-effective alternative for monitoring of *L. salmonis* in the field and to provide rapid notifications of potential infections to the farmers without causing welfare challenges for the fish. The results from this study exemplify the usefulness of the eDNA approach and the potential as an alternative to the standard monitoring practices. Nevertheless, a calibration needs to be established to transform eDNA sea lice gene copy numbers into sea lice individual count. To follow-up this research and its application to aquaculture monitoring, there is a need to confirm the observations made herein with a larger data set and over a longer period. Generally, if environmental DNA is to become a supplementary or alternative approach in sea lice assessment and monitoring for salmon farming, several challenges need to be investigated further. A promising avenue is also the implementation of such approach for real-time automation of sea lice detection such as with the ESP instrument. The present research offers very promising perspectives for the application of eDNA to aquaculture challenges.

## Supporting information

**S1 Appendix. Evaluation of the content of amplicons generated using SL2 assay.**
(DOCX)

**S1 Fig. Study area with the map with the localization of fish pen nets.**
(TIF)

**S2 Fig. Illustration of dissimilarities in the sequences between L. salmonids and *Caligus* sp.**
Fragments of the alignment generated for the randomly picked 136 (from 256) L. salmonids mitochondrial 16S rRNA sequences with regions primers and probe of qPCR assay target (on yellow). The number on blue–the start position (including gaps) for the oligo binding. In addition, for L. salmonids number of mismatches are provided.
(TIF)

**S3 Fig. LOD determined for SL2 assay.** LOD was determined from dilution series, 8 replicates were amplified at concentration of 550000, 110000, 22000, 4400, 880, 176, 35.2, 7.04, 1.408, 0.2816, 0.05632 and O copies $1\mu l^{-1}$. The proportion of positive amplifications are plotted against the standard concentrations (x- axis logarithmic). LOD was determined as the

minimum concentration of 95% replicates amplified (95% threshold is shown as a line).
(TIF)

**S4 Fig. LOQ-Limit of quantification for SL2 assay.** LOQ was determined from dilution series, 8 replicates were amplified at concentration of 550000, 110000, 22000, 4400, 880, 176, 35.2, 7.04, 1.408, 0.2816, 0.05632 1μl$^{-1}$. The coefficient of variation (relative standard deviation) (CV = 100$^*$(SD/mean)) was plotted against logarithmic transformed concentrations.
(TIF)

**S5 Fig. The number of DNA copies per 1L in the control tank and in the experimental tanks with infested fish.**
(TIF)

**S6 Fig. Distribution of estimated *L. salmonids* MGC at individual depths from samples collected in May and September.** N is number of samples included.
(TIF)

**S7 Fig. The comparison of SL2 qPCR assay performance on the ESP vs. bench top analyses on the StepOnePlus (SOPa) instrument.** Y-axis -Ct value, X-axis -gene copy number per 1 μL.
(TIF)

**S8 Fig. Number of L. salmonids found on fish determined by visual enumeration performed by a fish farmer following the national rules in Norway.**
(TIF)

**S1 Table. Description of samples collected from Oslofjord.**
(PPTX)

**S2 Table. Primers and probes targeted on L. salmonis used in the present study.**
(PPTX)

## Acknowledgments

We would like to acknowledge the project partners and at NVI David Strand and Trude Vrålstad, for their constructive discussions. We would also like to acknowledge the valuable contribution to field work made by Alan Le Tressoler (NORCE) and Britta Pedersen for laboratory assistance (DTU Aqua).

## Author Contributions

**Conceptualization:** Adriana Krolicka, Mari Mæland Nilsen, Thierry Baussant.

**Formal analysis:** Adriana Krolicka, Mari Mæland Nilsen.

**Funding acquisition:** Mari Mæland Nilsen, Fiona Provan, Thierry Baussant.

**Investigation:** Adriana Krolicka, Mari Mæland Nilsen, Brian Klitgaard Hansen, Magnus Wulf Jacobsen.

**Methodology:** Adriana Krolicka.

**Project administration:** Thierry Baussant.

**Supervision:** Thierry Baussant.

**Visualization:** Adriana Krolicka.

**Writing – original draft:** Adriana Krolicka, Mari Mæland Nilsen.

**Writing – review & editing:** Adriana Krolicka, Brian Klitgaard Hansen, Magnus Wulf Jacobsen, Fiona Provan, Thierry Baussant.

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
