## [Decision Letter · Decision Letter 0]

24 May 2022

PONE-D-21-27570Sea lice (Lepeophtherius salmonis) detection and quantification around aquaculture installations using quantification around aquaculture installations usingPLOS ONE

Dear Dr. Krolicka,

Thank you for submitting your manuscript to PLOS ONE. After careful consideration, we feel that it has merit but does not fully meet PLOS ONE’s publication criteria as it currently stands. Therefore, we invite you to submit a revised version of the manuscript that addresses the points raised during the review process.

Many thanks for submitting your manuscript to PLOS One

It was reviewed by two experts in the field, and they have recommended some modifications be made prior to acceptance

I therefore invite you to make these changes and to write a response to reviewers which will expedite revision upon resubmission

I wish you the best of luck with your modifications

Hope you are keeping safe and well in these difficult times

Thanks

Simon

We look forward to receiving your revised manuscript.

Kind regards,

Simon Clegg, PhD

Academic Editor

PLOS ONE

Journal Requirements:

2. In your Methods section, please include a comment about the state of the fish following this research. Were they euthanized or housed for use in further research? If any animals were sacrificed by the authors, please include the method of euthanasia and describe any efforts that were undertaken to reduce animal suffering.

3. We note that you are reporting an analysis of a microarray, next-generation sequencing, or deep sequencing data set. PLOS requires that authors comply with field-specific standards for preparation, recording, and deposition of data in repositories appropriate to their field. Please upload these data to a stable, public repository (such as ArrayExpress, Gene Expression Omnibus (GEO), DNA Data Bank of Japan (DDBJ), NCBI GenBank, NCBI Sequence Read Archive, or EMBL Nucleotide Sequence Database (ENA)). In your revised cover letter, please provide the relevant accession numbers that may be used to access these data. For a full list of recommended repositories, see http://journals.plos.org/plosone/s/data-availability#loc-omics or http://journals.plos.org/plosone/s/data-availability#loc-sequencing.

4. Please update your submission to use the PLOS LaTeX template. The template and more information on our requirements for LaTeX submissions can be found at http://journals.plos.org/plosone/s/latex.

7. Please amend either the title on the online submission form (via Edit Submission) or the title in the manuscript so that they are identical.

Reviewers' comments:

Reviewer's Responses to Questions

**Comments to the Author**

1. Is the manuscript technically sound, and do the data support the conclusions?

Reviewer #1: Yes

Reviewer #2: Yes

2. Has the statistical analysis been performed appropriately and rigorously? 

Reviewer #1: Yes

Reviewer #2: Yes

3. Have the authors made all data underlying the findings in their manuscript fully available?

Reviewer #1: Yes

Reviewer #2: Yes

4. Is the manuscript presented in an intelligible fashion and written in standard English?

Reviewer #1: Yes

Reviewer #2: Yes

5. Review Comments to the Author

Reviewer #1: Krolicka et al. have developed a new assay to specifically detect and quantify sea lice eDNA. The manuscript is well written.

There are few minor points.

Line 43: control aquaculture related diseases

Line 82: Italicize “L. salmonis”

Line 178-185: Can you explain why the L. salmonis naupli or copepodites were directly spiked to the filter rather than spiking them to seawater and conducting filtration afterwords.

Figure 1: Legends are too small

Line 332: “The numbers of MGC…” sentence may be not necessary.

Line 336: as 136

Line 369: production limiting factors

Line 408: to some extent

Line 429-431: The idea of this sentence

Line 467-473: Have you evaluated the temporal variation of L. salmonis eDNA?

Supplementary Figure 2: Pacific

Supplementary Figure 3: ….and 0 copies per µl

Supplementary Figure 5 and 6: Please show the statistical significance

Reviewer #2: The author presents a novel and important validation for an eDNA-based approach to sea lice monitoring. Given the ongoing issue of sea lice infestation on Norwegian aquaculture sites and the elusive nature of unattached infective stages, this may provide a powerful tool for early detection of present/future infestation pressure. Overall, this is an interesting and comprehensive study addressing a question of great ecological and economic importance in Norway. I outline a few specific points below that I hope the author will try to address if relevant to the interpretation of their results.

In the case of samples collected from aquaculture sites, is it possible to discriminate between eDNA from nauplii/copepodite stages and eDNA shed from attached or dead adult/sub-adult lice? Given the potentially large contribution of DNA from these larger lice stages, the presence of high numbers of adult/sub-adult lice may obscure the interpretation of the abundance of infective stages. It may be valuable to discuss briefly the potential contribution of adult/sub-adult DNA from farm samples if the author can estimate from their collected data if they believe this may be relevant to the interpretation of their results.

Lines 46-47: The author outlines their rationale for focusing on L. salmonis, as this is the primary sea lice species impacting Norwegian salmon farms. I am wondering if the author suspects this method could be applied to Caligus spp. Do you think special considerations would be required to apply such a method to Caligus spp. due to their broader host range or differences in lifecycle? Or do you think a similar eDNA-based monitoring method could be developed for members of this sea lice genus using a similar approach to that outlined in this this study?

Lines 63-65: As the author describes, L. salmonis exhibit non-parasitic stages. I was wondering if the author has thoughts on how one might incorporate such information into such an eDNA-based monitoring program and/or if it would be necessary for the effectiveness of this monitoring approach. For example, is it possible that eDNA from these non-parasitic stages could be detected from a nearby farm source but that did not necessarily translate to an elevated infection risk for the focal farm (given the latent period prior to these free-living stages becoming infective)? It sounds like the primary proposed application of this methodology is to assess increases in infestation pressure on fish farms, in which case the presence of non-infective stages would likely be related to the abundance of adult lice at a given site. If the author has considered if/how the presence of these non-infective stages may be incorporated into the application of an eDNA-based monitoring program, it may be worth briefly discussing.

Lines 390-400: The authors discuss ecological explanations for observed differences in depth stratified eDNA detections between sampling seasons. It may be worth discussing potential explanations for this phenomenon related to eDNA dispersal and/or stability. For example, is it possible that depth-dependent eDNA mixing and/or degradation rates were more variable in September than in May, independent of sea lice distributions?

Line 532: "Pacyfic" should be spelled Pacific

6. PLOS authors have the option to publish the peer review history of their article (what does this mean?). If published, this will include your full peer review and any attached files.

Reviewer #1: No

---

## [Author Response · Author response to Decision Letter 0]

10 Aug 2022

1. Responses to the reviewers 

We would like to thank both reviewers for the valuable comments. The comments raised by the reviewers brought our attention to some of the important but less visible aspects of the manuscript regarding methodology, eDNA based detection and quantification of sea lice. Below our answers to each point raised by the reviewers: 

Reviewer #1: Krolicka et al. have developed a new assay to specifically detect and quantify sea lice eDNA. The manuscript is well written.

There are few minor points.

Line 43: control aquaculture related diseases - has been corrected 

Line 82: Italicize “L. salmonis” -has been corrected

Line 178-185: Can you explain why the L. salmonis naupli or copepodites were directly spiked to the filter rather than spiking them to seawater and conducting filtration afterwords. 

We consider this comment being highly justified. The decision of spiking directly to the filter has been made because the overarching goal was to establish possible relationship between an exact number of individuals for two different life-stages and gene copy number/quantification. Spiking seawater with sea lice would be source of possible variation and uncertainties, as the small larvae tend to be retained on the walls of the glass bottles and not be filtered, with the risk that the relationship between the sea lice life-stages in seawater and the eDNA quantification could be biased. In the field, we are aware that eDNA may arise also from free DNA in addition to bulk individuals, but the intention was here to obtain an empirical relationship between number of individual and the measured gene copy numbers and ignoring the potential contribution of free DNA. 

Figure 1: Legends are too small -has been corrected 

Line 332: “The numbers of MGC…” sentence may be not necessary. -the sentence has been deleted 

Line 336: as 136 -has been corrected

Line 369: production limiting factors -has been corrected

Line 408: to some extent -has been corrected

Line 429-431: The idea of this sentence – the sentence has been modified 

Line 467-473: Have you evaluated the temporal variation of L. salmonis eDNA?

Supplementary Figure 2: Pacific- has been corrected

Supplementary Figure 3: ….and 0 copies per µl -has been corrected

Supplementary Figure 5 and 6: Please show the statistical significance – this has been done

Reviewer #2: The author presents a novel and important validation for an eDNA-based approach to sea lice monitoring. Given the ongoing issue of sea lice infestation on Norwegian aquaculture sites and the elusive nature of unattached infective stages, this may provide a powerful tool for early detection of present/future infestation pressure. Overall, this is an interesting and comprehensive study addressing a question of great ecological and economic importance in Norway. I outline a few specific points below that I hope the author will try to address if relevant to the interpretation of their results.

In the case of samples collected from aquaculture sites, is it possible to discriminate between eDNA from nauplii/copepodite stages and eDNA shed from attached or dead adult/sub-adult lice? Given the potentially large contribution of DNA from these larger lice stages, the presence of high numbers of adult/sub-adult lice may obscure the interpretation of the abundance of infective stages. It may be valuable to discuss briefly the potential contribution of adult/sub-adult DNA from farm samples if the author can estimate from their collected data if they believe this may be relevant to the interpretation of their results.

The reviewer is touching upon a tricky aspect, which is general for the environmental DNA methodology. With that approach, we cannot reliably discriminate nauplii/copepodite stages from attached or dead adult/sub-adult lice. Basically, we have too little knowledge and data to discuss in the manuscript when it comes to “the potentially large contribution of DNA from these larger lice stages, the presence of high numbers of adult/sub-adult lice may obscure the interpretation of the abundance of infective stages”. Although we believe that there will not be a large disproportion if comes to these numbers because as the reviewer mentioned eDNA from the attached stages will be mainly the fraction of shed eDNA, meanwhile free-living stages will be found in water as whole organisms or as their tissue fragments. This means we may in some instances overestimate the gene copy numbers for free living stages i.e. potential for the high infestation pressure. eRNA based detection could overcome the issue of detection and quantification of dead/living organisms and possibly the fraction of larval/adult eDNA. However, the sensor device Environmental Sampling Processor (ESP) with implemented PCR module used here is currently only able only to quantify DNA gene targets. In the manuscript we were trying to underline that long lasting eDNA based investigation through several seasons coupled with the data gained by fish farmers about numbers of sea lice counted on fish could provide information about fluctuations, possibly pattern, and provide support for how the eDNA based data could be interpreted. 

Lines 46-47: The author outlines their rationale for focusing on L. salmonis, as this is the primary sea lice species impacting Norwegian salmon farms. I am wondering if the author suspects this method could be applied to Caligus spp. Do you think special considerations would be required to apply such a method to Caligus spp. due to their broader host range or differences in lifecycle? Or do you think a similar eDNA-based monitoring method could be developed for members of this sea lice genus using a similar approach to that outlined in this this study?

Yes, this method could be applied to Caligus spp. Although the current regulation focus is on L. salmonis, we believe it would be useful to use a combination of eDNA based monitoring of L. salmonis and Caligus sp. (using an assay solely targeting Caligus sp.). That would provide more comprehensive information about sea lice infestation and threat to fish farms. The results based on the parallel monitoring would distinguish these two, especially this could be valuable in the geographic localities with larger impact from Caligus sp. 

Lines 63-65: As the author describes, L. salmonis exhibit non-parasitic stages. I was wondering if the author has thoughts on how one might incorporate such information into such an eDNA-based monitoring program and/or if it would be necessary for the effectiveness of this monitoring approach. For example, is it possible that eDNA from these non-parasitic stages could be detected from a nearby farm source but that did not necessarily translate to an elevated infection risk for the focal farm (given the latent period prior to these free-living stages becoming infective)? It sounds like the primary proposed application of this methodology is to assess increases in infestation pressure on fish farms, in which case the presence of non-infective stages would likely be related to the abundance of adult lice at a given site. If the author has considered if/how the presence of these non-infective stages may be incorporated into the application of an eDNA-based monitoring program, it may be worth briefly discussing.

The response to this comment was partly made to the reviewer’s first comment. Some additional, longer lasting study is needed to understand the seasonal fluctuations of sea lice eDNA detection and quantifications, and how this influences the real risk of infection. This was not part of the scope of the project of this research due to the limited financial resources. We agree this evaluation could lead to a better understanding how this type of data could be used by fish farmers to take actions and be prepared. We believe that for some period both type of investigations, traditional monitoring and eDNA based will need to be performed in parallel and then the eDNA based information could be utilized with good confidence to replace/support the conventional method. Lines 390-400: The authors discuss ecological explanations for observed differences in depth stratified eDNA detections between sampling seasons. It may be worth discussing potential explanations for this phenomenon related to eDNA dispersal and/or stability. For example, is it possible that depth-dependent eDNA mixing and/or degradation rates were more variable in September than in May, independent of sea lice distributions?

We agree with the reviewer, therefore we have added a short text (below) to the manuscript about phenomenon related to eDNA dispersal and/or stability. We should not exclude the possibility that depth-dependent eDNA mixing and/or degradation rates were more variable in September than in May and this phenomenon independent or partially independent of sea lice distributions. 

“Since the spatial and temporal signal of eDNA is dependent heavily on the environment it is possible that vertical mixing of water masses and/or degradation rates was more important than sea lice distributions. Season, light conditions, and water temperature are among the most important factors that impact distribution and eDNA stability (Harrison et al., 2019). The light conditions were different in May than September, UV light intensity was stronger in May than in September, therefore it is possible for example that UV light impacted on faster eDNA decay in upper layer in May, therefore no significant differences in MGC between individual layers were observed. “

Line 532: "Pacyfic" should be spelled Pacific- has been corrected

2. Responses to the Editor

The manuscript has been edited according to the PLOSOne recommendations 

2. In your Methods section, please include a comment about the state of the fish following this research. Were they euthanized or housed for use in further research? If any animals were sacrificed by the authors, please include the method of euthanasia and describe any efforts that were undertaken to reduce animal suffering.

We have added a following sentence to the Methods section: ” Fish were housed for use in further research and animals were not sacrificed by the authors.”

3. We note that you are reporting an analysis of a microarray, next-generation sequencing, or deep sequencing data set. PLOS requires that authors comply with field-specific standards for preparation, recording, and deposition of data in repositories appropriate to their field. Please upload these data to a stable, public repository (such as ArrayExpress, Gene Expression Omnibus (GEO), DNA Data Bank of Japan (DDBJ), NCBI GenBank, NCBI Sequence Read Archive, or EMBL Nucleotide Sequence Database (ENA)). In your revised cover letter, please provide the relevant accession numbers that may be used to access these data. For a full list of recommended repositories, see http://journals.plos.org/plosone/s/data-availability#loc-omics or http://journals.plos.org/plosone/s/data-availability#loc-sequencing.

The result of amplicon high throughput sequencing leading to demonstration of SL2 assay specificity has been submitted to NCBI Sequence Read Archive. The number of submissions has been added to the manuscript (BioSample accession SAMN29936430) 

4. Please update your submission to use the PLOS LaTeX template. The template and more information on our requirements for LaTeX submissions can be found at http://journals.plos.org/plosone/s/latex.

The manuscript has been edited according to LaTeX template.

All important information regarding Data Availability can be found in the cover letter.

We have provided information about DOI number in the cover letter.

7. Please amend either the title on the online submission form (via Edit Submission) or the title in the manuscript so that they are identical.

Thank you for this insight. We have made the titles identical. 

The following citation was added to the main text: Harrison JB, Sunday JM, Rogers SM. Predicting the fate of eDNA in the environment and implications for studying biodiversity. Proceedings of the Royal Society B: Biological Sciences. 2019;286(1915):20191409.

This has been done

---

## [Editor Report · Decision Letter 1]

6 Sep 2022

Sea lice (Lepeophtherius salmonis) detection and quantification around aquaculture installations using environmental DNA

PONE-D-21-27570R1

Dear Dr. Krolicka,

We’re pleased to inform you that your manuscript has been judged scientifically suitable for publication and will be formally accepted for publication once it meets all outstanding technical requirements.

Kind regards,

Simon Clegg, PhD

Academic Editor

PLOS ONE

Additional Editor Comments:

Many thanks for resubmitting your manuscript to PLOS One

As you have addressed all the comments and the manuscript reads well, I have recommended it for publication

You should hear from the Editorial Office shortly.

It was a pleasure working with you and I wish you the best of luck for your future research

Thanks

Simon

---

## [Editor Report · Acceptance letter]

12 Sep 2022

PONE-D-21-27570R1 

Sea lice *(Lepeophtherius salmonis)* detection and quantification around aquaculture installations using environmental DNA 

Dear Dr. Krolicka:

I'm pleased to inform you that your manuscript has been deemed suitable for publication in PLOS ONE. Congratulations! Your manuscript is now with our production department. 

Kind regards, 

on behalf of

Dr. Simon Clegg 

Academic Editor

PLOS ONE